# Pregnant Women’s Knowledge of and Attitudes towards Influenza Vaccination during the COVID-19 Pandemic in Poland

**DOI:** 10.3390/ijerph19084504

**Published:** 2022-04-08

**Authors:** Agata Pisula, Agnieszka Sienicka, Katarzyna Karina Pawlik, Agnieszka Dobrowolska-Redo, Joanna Kacperczyk-Bartnik, Ewa Romejko-Wolniewicz

**Affiliations:** 1Students’ Scientific Group Affiliated to 2nd Department of Obstetrics and Gynecology, Medical University of Warsaw, 00-315 Warsaw, Poland; sienicka.aa@gmail.com (A.S.); katarzynapawlik97@gmail.com (K.K.P.); 22nd Department of Obstetrics and Gynecology, Medical University of Warsaw, 00-315 Warsaw, Poland; agnieszka.dobrowolskaredo@gmail.com (A.D.-R.); asiakacperczyk@gmail.com (J.K.-B.); ewamariar@wp.pl (E.R.-W.)

**Keywords:** influenza, COVID-19, pandemic, vaccination

## Abstract

Pregnant women are more susceptible to influenza virus infections due to the immunological and physiological changes in the course of pregnancy. Vaccination during pregnancy is a safe and effective method for protecting both the mothers and the infants from influenza and its complications. This study was conducted in order to determine the knowledge and attitudes of Polish pregnant women towards influenza vaccination during the COVID-19 pandemic. A questionnaire-based and self-administered study was carried out fully online and a total of 515 women participated. A total of 52% (n = 268) of surveyed women answered that vaccination against influenza during pregnancy was safe. However, only 21% (n = 108) were vaccinated against influenza during their current pregnancy and 17.5% (n = 90) intended to be vaccinated. The participants indicated many concerns about getting vaccinated during pregnancy, but also many benefits that come with the vaccination. General knowledge about influenza, its complications, and vaccination was quite high in the study group.

## 1. Introduction

Pregnancy is a unique immunological condition that protects the fetus from maternal rejection, while also allowing fetal development and providing protection against pathogens [1]. Viral infections during pregnancy are associated with a risk of obstetric complications, acute severe maternal illness, vertical transmission to the fetus, and perinatal transmission during delivery [2]. Many hormones (estrogen, progesterone) and factors, whose levels increase during pregnancy, have immunosuppressive effects. Changes in immune system function and changes in hormone production make pregnant women far more susceptible to infections. Pregnant women are at higher risk of severe complications and mortality from many viral diseases, including influenza [3]. In Poland, as well as in most European countries, vaccination is recommended for all women planning to become pregnant or who are pregnant [4,5]. Vaccination against influenza effectively prevents illness. Vaccines may be administered at any time during pregnancy, optimally in the second or third trimester. Vaccination with the inactivated vaccine is safe for both the mother and the child, and reduces the risk of illness in newborns in the first six months of life. It is especially beneficial since a child at this age may not yet be vaccinated against influenza [6]. The COVID-19 pandemic has caused much fear and concern among people due to the limited access to healthcare systems as well as loss of income. It has led to an increased interest in infectious diseases and prevention, but also to a change in reproductive intentions. Many couples have postponed their plans to have children, reduced the number of children they initially planned to have, or decided not to have children at all [7]. On 27 December 2020, the vaccination program against COVID-19 began in Poland, and now COVID-19 vaccination is also recommended for pregnant women [8]. The vaccination program and the development of an efficient healthcare system during the pandemic may have an impact on increasing the desire to have children, but this depends, despite other issues, on the knowledge and attitudes of pregnant women towards vaccinations. The aim of our study was to determine the knowledge and attitudes of Polish pregnant women towards influenza vaccination during the COVID-19 pandemic.

## 2. Materials and Methods

A questionnaire-based study was carried out in order to elicit and analyze the knowledge of and attitudes towards influenza vaccination amongst pregnant women in Poland during the COVID-19 pandemic. The self-administered and voluntary questionnaire was created online using the survey administration software Google Forms and distributed on 92 Polish Facebook groups dedicated to women, mothers, or pregnant women. The participants were random Polish pregnant women who were members of the social media groups. Anonymity and confidentiality were ensured. The data were collected for over 2 weeks, from 24 October to 9 November 2021. 

The questionnaire included a total of 50 questions, both single- and multiple-choice questions, and was divided into several sections. The first part gathered basic sociodemographic and economic data including age, educational level, marital status, place of residence, and an average income per household member. The second part involved the participant’s obstetric history and details of the current pregnancy. The information regarding women’s knowledge about influenza and its vaccine as well as factors influencing vaccination and non-vaccination were also collected. The participants were also asked about their vaccination status against both influenza and COVID-19 together with their actual and preferred sources of information on the vaccination. The entire questionnaire is available in Appendix A.

With the purpose of achieving a 95% confidence level and a 5% margin of error, the sample size of the population of pregnant women was calculated to be 385 participants. 

All questionnaires were completed correctly. Obtained data were analyzed using descriptive statistics in Microsoft Excel and the univariate and multivariate logistic regression analyses for the categorical variables. Statistical significance was detected by a *p*-value less than 0.05.

## 3. Results

A total of 515 women participated in the study. All of the answers were completed correctly and used for further analysis. The baseline sociodemographic features are described in Table 1.

Amongst all of the respondents, 196 (38.1%) answered that they have had influenza; however, only 34 (6.6%) had an influenza test performed. Less than half (n = 224, 43.5%) had been vaccinated against influenza and only 108 (21%) had been vaccinated during their current pregnancy (Table 2). 

In the univariate regression analysis, a correlation between influenza vaccination status and age was found in the 31–40 age group. Significant association was also found between the vaccination status and the average income per household member and with the place of residence. In the multivariate analysis, the predictor that influenced the incidence of influenza vaccination was the average income per family member. No significant relationships between influenza vaccination and education or relationship status were found; however, this might be due to an uneven distribution of the participants in those groups (Table 3 and Table 4). The distribution of the results regarding participant’s vaccination status against influenza and the sociodemographic characteristics is shown in Appendix A.

Regarding the status of influenza vaccination in their current pregnancy, both in the univariate and multivariate logistic regression analyses, average income per household member and place of residence were the factors that influenced the decision to get vaccinated (Table 5 and Table 6). The distribution of the results regarding participant’s vaccination status against influenza during their current pregnancy and the sociodemographic characteristics is shown in Appendix A.

The vast majority knew that influenza is a virus-transmitted disease (n = 497, 96.5%). A summary of the distribution of answers to the questions on knowledge about risk groups for severe influenza and influenza treatment is presented in Figure 1 and Figure 2.

The majority of women had heard about influenza complications that can occur during pregnancy and in infants. The most commonly indicated complications of influenza in pregnancy were pneumonia (n = 265, 51.5%), premature delivery (n = 237, 46%), and acute respiratory failure (n = 233, 45.2%). Among the post-influenza complications in newborns, pneumonia (n = 243, 47.2%), respiratory failure (n = 215, 41.7%), and myocarditis (n = 197, 38.3%) were the most frequently chosen. A total of 141 (27.4%) participants had not heard of any complications of influenza in pregnancy and 195 (37.9%) had not heard of any complications of influenza in infants (Table 7 and Table 8).

A total of 52% (n = 268) of surveyed women believed that vaccination against influenza during pregnancy is safe. Despite this, only 21% (n = 108) had been vaccinated against influenza in their current pregnancy and 17.5% (n = 90) intended to be vaccinated. A total of 15.3% (n = 79) did not know yet if they will get vaccinated and 46.2% (n = 238) did not intend to be vaccinated. 

The most common reasons for being vaccinated against influenza during pregnancy chosen by vaccinated women (n = 108) was a desire to protect themselves from infection (n = 103, 95.4%) and to protect the fetus (n = 103, 95.4%). A total of 28 (25.9%) women also indicated that they got vaccinated because of a doctor’s recommendation. Out of the 108 women who reported they were vaccinated against influenza during their current pregnancy, 8 (7.4%) received the vaccination in the first trimester, 51 (47.2%) in the second trimester, and 49 (45.4%) in the third trimester.

Out of the women who had not been vaccinated during their current pregnancy and did not intend to be vaccinated (n = 238), 131 (55%) avoided taking medications or medicinal products during pregnancy, 128 (53.8%) were concerned about the impact of vaccination on the fetus, 114 (47.9%) were concerned about potential post-vaccination complications, 71 (29.8%) believed that there is a small risk that they would contract influenza, 59 (24.8%) thought that the vaccination is ineffective, 36 (15.1%) indicated bad experiences with previous vaccinations, and 26 (10.9%) considered influenza not to be a dangerous illness. A total of 64 (26.9%) participants indicated other reasons.

Among women who had not yet been vaccinated but intend to do so (n = 90), 25 (27.8%) preferred to wait with the vaccination until an advanced stage of pregnancy, while 21 (23.3%) were advised by their doctor to be vaccinated later. A total of 15 (16.7%) women indicated vaccine unavailability and 10 (11.1%) lack of time as a reason. Ten (11.1%) women were not aware of a facility where they can be vaccinated, nine (10%) had not yet received sufficient information regarding the vaccination, and eight (8.9%) were waiting for the vaccination appointment. A total of 29 (32.2%) responders indicated other reasons. A total of 48 (53.3%) women planned to get vaccinated in the second trimester, 39 (43.3%) in the third trimester, and 3 (3.3%) after delivery. Of the 17 respondents who were in first trimester, 16 (94.1%) planned to vaccinate in the second trimester, and 1 (5.9%) in third trimester. Among the 41 women in the second trimester, 32 (78%) claimed they will vaccinate in the second and 9 (22%) in the third. Taking 32 pregnant women in the third trimester, 29 (90.6%) will vaccinate in the third trimester and 3 (9.4%) will vaccinate after delivery.

A total of 15.3% (n = 79) of the participants did not yet know if they will vaccinate and the main reasons for not vaccinating yet were: fear of potential vaccine complications (n = 29, 36.7%), concern about the effects of vaccination on the fetus (n = 29, 36.7%), avoiding medications or medicinal products during pregnancy (n = 26, 32.9%), small risk of contracting influenza (n = 23, 29.1%), unawareness of vaccination opportunity (n = 20, 25.3%), bad experiences with previous vaccinations (n = 10, 12.7%), and others (n = 31, 39.2%).

In our survey, the participants were also asked how had they obtained information about influenza and influenza vaccination, and how they would prefer to gain such knowledge. Internet and social media were indicated as the main sources of information (n = 192, 37.3%), while 43.3% (n = 223) of participants had not been searching for such information at all. A total of 81.2% (n = 418) of women stated they would like to receive details about influenza and influenza vaccination during pregnancy from a gynecologist, 38.6% (n = 199) from a general practitioner, and 35.5% (n = 183) from a midwife or a nurse. However, only for 20.6% (n = 106) of the women, a gynecologist was the actual source of information and only 27.6% (n = 142) were offered influenza vaccination by their gynecologist. Most participants were not offered to be vaccinated against influenza by anyone, neither before the current pregnancy (n = 292, 56.7%) nor during the current pregnancy (n = 309, 60%). 

Most participants correctly selected the vaccinations recommended for pregnant women in Poland: against influenza (n = 367, 71.3%) and against pertussis (n = 385, 74.8%). A total of 11.7% (n = 60) did not know which vaccinations are recommended and 5.6% (n = 29) thought that no vaccinations are recommended during pregnancy.

The subjective estimation of the general knowledge about influenza and influenza vaccination was quite high in the studied population. Most participants assessed their level of knowledge as sufficient to make a conscious decision about this vaccination (n = 319, 61.9%).

Our study also collected information about pregnant women’s knowledge and attitude to COVID-19 vaccination. 

The majority (n = 353, 68.5%) had not had COVID-19, with 28.5% (n = 147) having COVID-19 before pregnancy and 2.9% (n = 15) during pregnancy.

The safety of the COVID-19 vaccine in pregnancy was rated by participants similarly to the safety of the influenza vaccine. A total of 44.7% (n = 230) of women answered that COVID-19 vaccination is safe during pregnancy. A total of 38.3% (n = 197) of women were offered COVID-19 vaccination during pregnancy by their gynecologist, while approximately half of the women (n = 260, 50.5%) were not offered this vaccination during pregnancy by anyone. A total of 81.6% (n = 420) of the respondents would like to obtain the information about COVID-19 vaccination during pregnancy from a gynecologist, 47.6% (n = 245) from a general practitioner, and 34.8% (n = 179) from a midwife or a nurse.

A total of 154 (29.9%) respondents had been vaccinated against COVID-19 before pregnancy and 145 (28.2%) during pregnancy: 23 (15.9%) in the first trimester, 103 (71%) in the second trimester, and 19 (13.1%) in the third trimester. As the reasons for vaccinating, the women indicated the desire to protect themselves (n = 273, 91.3%), to protect the fetus (n = 153, 51.2%), doctor’s recommendation of the vaccine (n = 105, 35.1%), the desire to make travelling easier (n = 13, 4.3%), and to protect other people (n = 12, 4%). Twelve (4%) women indicated other reasons. 

In the univariate regression analysis, a significant association was found between the COVID-19 vaccination status and variables such as the educational status, average income per household member, and the place of residence. The multivariate analysis also showed a correlation between the vaccination and average income and place of residence. No significant relationship between COVID-19 vaccination and age or relationship status was found (Table 9 and Table 10). The distribution of the results regarding participant’s vaccination status against COVID-19 and the sociodemographic characteristics is shown in Appendix A.

A total 23.2% (n = 120) of women did not intend to be vaccinated against COVID-19, 12.4% (n = 64) did not know yet if they will vaccinate, and 6.2% (n = 32) had not yet been vaccinated but intend to do so: 1 (3.1%) participant would like to vaccinate in the second trimester, 3 (9.4%) in the third trimester, and 28 (87.5%) after the delivery.

Among the women who did not intend to be vaccinated (n = 120), 104 (86.7%) were concerned about the impact of vaccination on the fetus, 103 (85.8%) were concerned about potential post-vaccination complications, 67 (55.8%) avoided taking medications or medicinal products during pregnancy, 59 (49.2%) believed that vaccination is ineffective, 24 (20%) considered COVID-19 as not a dangerous illness, 19 (15.8%) believed that there is a small risk that they would contract COVID-19, 18 (15%) had bad experiences with previous vaccinations, 14 (11.7%) thought that the vaccine is not yet thoroughly examined, and 14 (11.7%) participants indicated other reasons.

There is a visible association between COVID-19 and influenza vaccine uptake during pregnancy. Women who had been vaccinated against COVID-19 were more likely to take the influenza vaccine and women who had not been vaccinated against COVID-19 also had not intended to vaccinate against influenza (Table 11).

## 4. Discussion

Pregnant women are more vulnerable to influenza virus infections as a result of the immunological and physiological changes that occur during pregnancy [9]. Vaccination during pregnancy is a safe and effective method for protecting both the mothers and the infants from influenza and its complications [10,11]. Influenza vaccination is even more important now during the COVID-19 pandemic. Studies have shown that people vaccinated against influenza are less likely to become infected with SARS-CoV-2 and have better clinical outcomes when they contract COVID-19 [12,13,14].

To the best of our knowledge, this is the first study to assess the uptake of influenza vaccination and knowledge of influenza and its complications amongst pregnant women in Poland in the context of the COVID-19 pandemic. We conducted a cross-sectional online study in order to elicit and analyze the knowledge and attitudes towards the influenza vaccination amongst pregnant women in Poland during the COVID-19 pandemic period. Our survey found that even though the general awareness of influenza and its complications both in pregnant women and infants were overall high and more than half believed that vaccination against influenza during pregnancy is safe, less than 40% of respondents had been vaccinated or intended to be vaccinated against influenza in their current pregnancy.

As interest in influenza vaccination during a pandemic increases, so does the number of studies on women’s attitudes towards influenza vaccination. A multi-center cross-sectional survey was recently conducted in China to evaluate influenza vaccination acceptance and associated factors among pregnant women during the COVID-19 pandemic. The total acceptance rate was 76.5%; however, many participants declared concerns about influenza vaccination during pregnancy. The most frequently mentioned ones were the worry about side effects and the influence of the vaccination on the fetus [15]. Our study also showed that pregnant women in Poland have many concerns about influenza vaccination during pregnancy. Over half of the pregnant women who had not got vaccinated also indicated the fear of the impact of vaccination on the fetus as the main reason. This fear of adverse events seems to be an important factor, since it was already the prevalent reason for refusing influenza vaccination in the study conducted in Greece in 2018 [16].

Although there is still a low influenza vaccination rate among pregnant women in Poland, there seems to be an increasing vaccination trend in comparison to previous studies carried out in Poland regarding the same topic. In a similar study conducted in Szczecin, Poland, in the 2019/2020 season among pregnant women, 5.9% of them reported receiving vaccination in the 2019/2020 season and 9.0% of them were planning to be vaccinated against influenza in the following season [17]. In another cross-sectional study conducted in Warsaw, Poland, in the 2017/2018 influenza season, only about 3% of women of childbearing age declared having been vaccinated, and a similar percentage of participants declared their willingness to get vaccinated against influenza during pregnancy in the upcoming influenza season [18]. 

This increase in vaccination acceptance shown in our study might be influenced by the pandemic and COVID-19 vaccination program, since there is an association between COVID-19 and influenza vaccine uptake amongst respondents. A similar increase in influenza vaccine uptake was shown in the study performed on the general population in the United States, where a 7.62% improvement was noticed [19]. Moreover, an interesting study was conducted in the United Kingdom, for which one of the objectives was to examine the impact of the COVID-19 pandemic on pregnant women’s attitudes towards maternal vaccines. Participants in the mentioned study believed that the pandemic had elevated the importance of routine maternal vaccines, which include the influenza vaccine [20]. This suggests that the higher influenza vaccine uptake noticeable in our study compared to previous studies might be influenced by the growing awareness of viral infections and the importance of vaccines due to the COVID-19 pandemic. 

Our study should be interpreted in light of a few limitations. Participants were recruited only if they were a member of women-, motherhood-, or pregnancy-oriented Facebook groups, so it may not represent Polish women in general. We do not have access to information on the number of pregnant women participating on these social networks. A total of 83.7% of respondents had higher education and this may also not be representative of Polish society. Furthermore, the survey was performed using a self-reported, cross-sectional web survey with a closed design, which might lead to some misinterpretations. The study was conducted entirely online and this may also affect the results. However, this form of data collection allowed us to gather information about pregnant women from different backgrounds and regions of the country.

## 5. Conclusions

Both influenza and COVID-19 can pose a serious threat to the health and life of the mother as well as the infant. Vaccination is an effective way to reduce the potential risk of complications caused by a disease. The majority of women in our study believed that vaccination against influenza during pregnancy is safe and the general awareness of influenza and its complications was rather high. Most of the women who did not intend to get vaccinated were concerned about the impact of vaccination on the fetus or avoided taking medications during pregnancy. The most common reason for being vaccinated against influenza during pregnancy chosen by vaccinated women was a desire to protect themselves and the fetus. Despite the fact that there is still a low influenza vaccination rate in pregnant women in Poland, when compared to other studies conducted in Poland previously, there is a visible increase in influenza vaccine uptake by pregnant women. Since there is an association between COVID-19 and influenza vaccine uptake amongst respondents, we might suspect that this increase in vaccination acceptance is influenced by the pandemic and the increase of awareness about the importance of vaccination.

## Figures and Tables

**Figure 1 ijerph-19-04504-f001:**
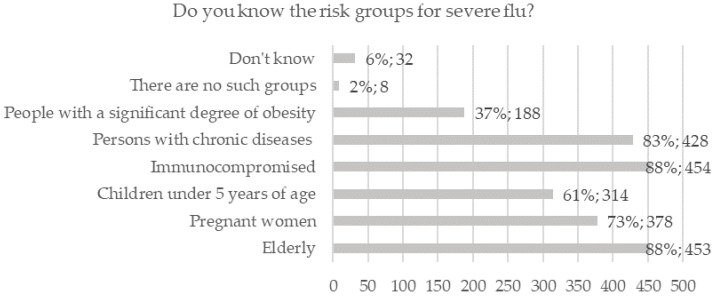
Respondents’ knowledge about risk groups for severe influenza.

**Figure 2 ijerph-19-04504-f002:**
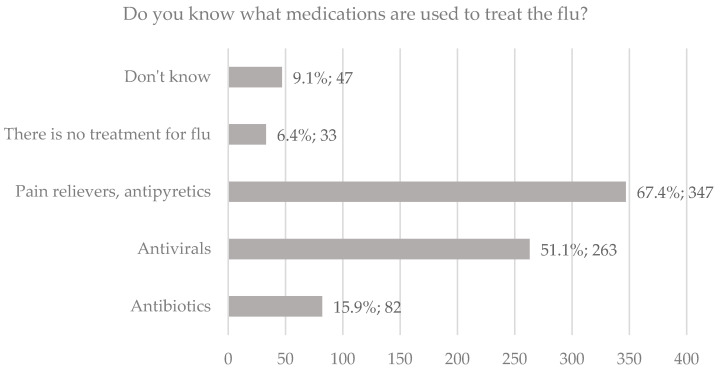
Respondents’ knowledge about influenza treatment.

**Table 1 ijerph-19-04504-t001:** Sociodemographic and obstetric characteristics of participants.

Category (n = 515)	Variables	Frequency	Percentage
Age	19–25	64	12.4
	26–30	223	43.3
	31–35	164	31.8
	36–40	57	11.1
	41–43	7	1.4
Education	Primary	2	0.4
	Vocational	5	1.0
	Secondary	59	11.5
	Univeristy students	18	3.5
	Higher	431	83.7
Average income per household member	<1000 PLN	15	2.9
	1000–2000 PLN	65	12.6
	2000–3000 PLN	108	21.0
	3000–4000 PLN	132	25.6
	4000–5000 PLN	79	15.3
	>5000 PLN	116	22.5
Place of residence	Countryside	104	20.2
	Small village (<50 k residents)	68	13.2
	Town (50–100 k residents)	34	6.6
	City (100–500 k)	101	19.6
	City (>500 k)	209	40.6
Current relationship status	Single	3	0.6
	Informal relationship	96	18.6
	Married	414	80.4
	Divorced	2	0.4
Week of gestation	1–13	60	11.7
	14–27	196	38.1
	28–40	259	50.3
Number of pregnancies	1	255	49.5
	2	167	32.4
	3	63	12.2
	4+	30	5.8
Number of labours	0	309	60.0
	1	154	29.9
	2	41	8.0
	3+	11	2.1
Pregnancy	Single	508	98.6
	Multiple	7	1.4
Having children	Yes	204	39.6
	No	311	60.4

**Table 2 ijerph-19-04504-t002:** History of influenza and influenza vaccination status.

Category (n = 515)	Variables	Frequency	Percentage
Have you ever had influenza?	Yes	196	38.1
	No	146	28.3
	I don’t know	173	33.6
Have you ever had an influenza test performed?	Yes	34	6.6
	No	464	90.1
	I don’t know	17	3.3
Have you ever been vaccinated against influenza?	Yes	224	43.5
	No	291	56.5
Have you been vaccinated against influenza during current pregnancy?	Yes	108	21.0
	No, but I am going to	90	17.5
	No and I am not going to	238	46.2
	No and I do not know if I will get vaccinated	79	15.3

**Table 3 ijerph-19-04504-t003:** Univariate regression analysis regarding influenza vaccination status.

Category (n = 515)				*p*-Value (LR Test)
Age				0.0374
Education				0.0743
Average income per household member				0.0001
Place of residence				0.0192
Current relationship status				0.3416
		95% CI		
Variables	OR	Lower	Upper	*p*-value
Age: 19–25 years old vs.				0.0374
26–30y	1.694	0.931	3.083	0.0843
31–35y	2.045	1.103	3.794	0.0232
36–40y	3.032	1.433	6.413	0.0037
41–43y	3.158	0.644	15.487	0.1564
Average income per household member: <1000 PLN vs.				0.0001
1000–2000 PLN	1.200	0.299	4.820	0.7972
2000–3000 PLN	2.353	0.626	8.844	0.2053
3000–4000 PLN	4.000	1.079	14.830	0.0381
4000–5000 PLN	3.349	0.877	12.795	0.0772
>5000 PLN	4.923	1.319	18.375	0.0177
Place of residence: Countryside vs.				0.0192
Small village (<50 k residents)	1.430	0.765	2.670	0.2622
Town (50–100 k residents)	0.652	0.276	1.542	0.3302
City (100–500 k)	1.289	0.734	2.265	0.3773
City (>500 k)	1.882	1.159	3.056	0.0106

**Table 4 ijerph-19-04504-t004:** Multivariate analysis regarding influenza vaccination status.

		95% CI		
Variables	OR	Lower	Upper	*p*-Value
Average income per household member: <1000 vs.				0.009431
1000–2000 PLN	1.400	0.339	5.791	0.642
2000–3000 PLN	2.397	0.617	9.306	0.206
3000–4000 PLN	3.949	1.030	15.146	0.045
4000–5000 PLN	2.960	0.742	11.816	0.124
>5000 PLN	4.386	1.130	17.031	0.033

**Table 5 ijerph-19-04504-t005:** Univariate regression analysis regarding influenza vaccination status during current pregnancy.

Category (n = 515)				*p*-Value (LR Test)
Age				0.1736
Education				0.1204
Average income per household member				0.0000
Place of residence				0.0000
Current relationship status				0.4046
		95% CI		
Variables	OR	Lower	Upper	*p*-value
Average income per household member: <1000 PLN vs.				0.0000
1000–2000 PLN	0.727	0.173	3.049	0.6632
2000–3000 PLN	1.610	0.425	6.102	0.4833
3000–4000 PLN	2.947	0.794	10.939	0.1062
4000–5000 PLN	3.022	0.790	11.557	0.1061
>5000 PLN	4.923	1.319	18.375	0.0177
Place of residence: Countryside vs.				0.0000
Small village (<50 k residents)	1.322	0.683	2.558	0.4076
Town (50–100 k residents)	0.162	0.036	0.718	0.0166
City (100–500 k)	1.255	0.691	2.280	0.4559
City (>500 k)	2.959	1.781	4.918	0.0000

**Table 6 ijerph-19-04504-t006:** Multivariate analysis regarding influenza vaccination status during current pregnancy.

		95% CI		
Variables	OR	Lower	Upper	*p*-Value
Average income per household member: <1000 PLN vs.				0.001022
>5000 PLN	3.896	1.001	15.170	0.050
Place of residence: Countryside vs.				0.000284
Town (50–100 k residents)	0.144	0.032	0.651	0.012
City (>500 k)	2.126	1.237	3.654	0.006

**Table 7 ijerph-19-04504-t007:** Participants’ knowledge about the influenza complications in pregnancy.

Complications (n = 515)	Frequency	Percentage
Pneumonia	265	51.5
Premature delivery	237	46.0
Acute respiratory failure	233	45.2
Cardiopulmonary arrest	195	37.9
Spontaneous abortion	172	33.4
Low birth weight	122	23.7
Intrauterine fetal demise	121	23.5
Fetal tachyarrhythmia	74	14.4
Have not heard about any complications	141	27.4

**Table 8 ijerph-19-04504-t008:** Participants’ knowledge about the influenza complications in infants.

Complications (n = 515)	Frequency	Percentage
Pneumonia	243	47.2
Respiratory failure	215	41.7
Myocarditis	197	38.3
Bronchitis	175	34.0
Meningitis	123	23.9
Sepsis	107	20.8
Have not heard about any complications	195	37.9

**Table 9 ijerph-19-04504-t009:** Univariate regression analysis regarding COVID-19 vaccination status.

Category (n = 515)				*p*-Value (LR Test)
Age				0.1281
Education				0.0001
Average income per household member				0.0000
Place of residence				0.0000
Current relationship status				0.2405
		95% CI		
Variables	OR	Lower	Upper	*p*-value
Education: Primary and Vocational vs.				0.0001
Secondary	1.714	0.307	9.575	0.5391
University Students	3.125	0.474	20.584	0.2361
Higher	5.423	1.039	28.304	0.0449
Average income per household member: <1000 PLN vs.				0.0000
1000–2000 PLN	1.506	0.430	5.268	0.5216
2000–3000 PLN	3.437	1.029	11.478	0.0447
3000–4000 PLN	6.558	1.967	21.858	0.0022
4000–5000 PLN	8.112	2.320	28.364	0.0010
>5000 PLN	10.524	3.083	36.044	0.0002
Place of residence: Countryside vs.				0.0000
Small village (<50 k residents)	1.510	0.815	2.794	0.1900
Town (50–100 k residents)	1.122	0.517	2.436	0.7701
City (100–500 k)	2.117	1.206	3.714	0.0090
City (>500 k)	3.741	2.264	6.183	0.0000

**Table 10 ijerph-19-04504-t010:** Multivariate analysis regarding COVID-19 vaccination status.

		95% CI		
Variables	OR	Lower	Upper	*p*-Value
Average income per household member: <1000 PLN vs.				0.000314
3000–4000 PLN	4.484	1.288	15.605	0.018
4000–5000 PLN	4.693	1.272	17.311	0.020
>5000 PLN	6.273	1.745	22.553	0.005
Place of residence: Countryside vs.				0.027920
City (>500 k)	2.407	1.396	4.148	0.002

**Table 11 ijerph-19-04504-t011:** Association between COVID-19 and influenza vaccine uptake during pregnancy.

	Vaccination against Influenza								
Vaccination against COVID-19	Yes		No, but I am going to		No and I am not going to		No and I do not know if I will get vaccinated		Spearman rank coefficient
	n	%	n	%	n	%	n	%	
Yes, before pregnancy	46	8.9	55	10.7	34	6.6	19	3.7	0.505952 (*p* = 0.0000)
Yes, during pregnancy	54	10.5	29	5.6	36	7.0	26	5.0	
No, but I am going to	3	0.6	1	0.2	22	4.3	6	1.2	
No and I am not going to	1	0.2	1	0.2	111	21.6	7	1.4	
No and I do not know if I will get vaccinated	4	0.8	4	0.8	35	6.8	21	4.1	

## Data Availability

All the data used in this study was presented in the article.

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
