# Peer review of "Pregnant Women’s Knowledge of and Attitudes towards Influenza Vaccination during the COVID-19 Pandemic in Poland"

_ijerph, 2022, doi:10.3390/ijerph19084504_

Round 1

Reviewer 1 Report

The study is novel as it provides insights into pregnant women's attitudes towards influenza vaccination under the global pandemic caused by SARS-CoV-2. The authors gathered and analyzed important information among pregnant women, which are especially vulnerable to infections. The results may help public health policymakers, medical practitioners, and educators improve vaccine acceptance.

  1. Including variables such as age, social-economic status, educational levels, etc of women in general in Poland will be helpful in interpreting the results. For example, 83.7% of survey takers had higher education. If this is not representative of Polish society, it is a limitation of the study and the results could be biased.
  2. The first four groups in educational levels combined are only 16.4% compared to "Higher", 87.3%. This should make lines 79-82 "No significant relationships between influenza vaccination and education or relationship status were found." a weak claim.
  3. Lines 88-94 - it would be nice to include a breakdown of variables, instead of percentages of all survey takers.
  4. Lines 112-114 - only 11.7% of the survey takers were in the 1st trimester, which inevitably contributes to the low percentage of women getting the vaccination in the 1st trimester. Similar comparisons should be provided for the 2nd and 3rd trimesters to prevent biased conclusions.
  5. Lines 129-130 - a breakdown of stages of pregnancy should be included.
  6. Line 158 - 68.5% isn't "vast majority".
  7. Lines 165-167 - adding a Venn diagram will be very helpful in interpreting the results.
  8. Lines 184-191 - breakdown of variables such as social-economic status and education levels will help interpret the attitude of not intending to be vaccinated.
  9. Line 185 - typo "also have not intended to vaccinate against influenza"
  10. Lines 247-253 - internet access is another limitation and should be added
  11. Lines 255-262 - conclusions should shed more light on the current study, instead of known facts.

Author Response

Response to Reviewer 1 Comments

Point 1:  Including variables such as age, social-economic status, educational levels, etc of women in general in Poland will be helpful in interpreting the results. For example, 83.7% of survey takers had higher education. If this is not representative of Polish society, it is a limitation of the study and the results could be biased.

Response 1:  In our study, such variables were considered and also described separately in tables. Indeed, 83.7% of respondents had higher education and it is not representative of Polish society. We have included this information as a limitation of the study.

Point 2:  The first four groups in educational levels combined are only 16.4% compared to "Higher", 87.3%. This should make lines 79-82 "No significant relationships between influenza vaccination and education or relationship status were found." a weak claim.

Response 2: We included the fact of inequality in the distribution of respondents in this group in our claim.

Point 3: Lines 88-94 - it would be nice to include a breakdown of variables, instead of percentages of all survey takers.

Response 3: Thanks for the suggestion, we changed the form of presenting these results into figures.

Point 4: Lines 112-114 - only 11.7% of the survey takers were in the 1st trimester, which inevitably contributes to the low percentage of women getting the vaccination in the 1st trimester. Similar comparisons should be provided for the 2nd and 3rd trimesters to prevent biased conclusions.

Response 4:  We understand your point, so we have included the full number of pregnant women who declared vaccination in their current pregnancy. However, the purpose of this question was to examine trends in the timing of vaccination. Women in the second and third trimesters of pregnancy had the opportunity to be vaccinated in the first trimester but chose to be vaccinated later, so the low number of women vaccinated in the first trimester is not due to the low number of first trimester respondents.

Point 5: Lines 129-130 - a breakdown of stages of pregnancy should be included.

Response 5: Thank you for your advice, we included a breakdown of stages of pregnancy.

Point 6: Line 158 - 68.5% isn't "vast majority".

Response 6: Thank you, we have made a correction.

Point 7: Lines 165-167 - adding a Venn diagram will be very helpful in interpreting the results.

Response 7: Thank you very much for your suggestion. In our opinion a Venn diagram will not be appropriate for such data.

Point 8: Lines 184-191 - breakdown of variables such as social-economic status and education levels will help interpret the attitude of not intending to be vaccinated.

Response 8: We carefully evaluated all variables and correlations and they were not significantly different from the socio-demographic data presented in Table 1.

Point 9: Line 185 - typo "also have not intended to vaccinate against influenza"

Response 9: Thank you, we have made a correction.

Point 10: Lines 247-253 - internet access is another limitation and should be added

Response 10: Thank you, we have made a correction.

Point 11: Lines 255-262 - conclusions should shed more light on the current study, instead of known facts.

Response 11: Thank you, we have made corrections as suggested.

Reviewer 2 Report

  1. Methods: line 57: please correct to “pregnant women” instead of “pregnancy”.  

  1. Methods: Do you know (approximately) the number of women participating on these social networks? If yes, please indicate the overall response rate (%) in the first line of Results. If no, please add this to the limitations paragraph (Discussion).     

  1. Table 1: You mention “studying”: you mean “university students”? if yes, please correct accordingly.

  1. In Tables 3 and 4 there are significant associations with income. Please mention in Results section.

  1. Please add the following recently published articles in your reference list:

Vaccination programs for pregnant women in Europe, 2021. Maltezou HC, Effraimidou E, Cassimos DC, Medic S, Topalidou M, Konstantinidis T, Theodoridou M, Rodolakis A. Vaccine. 2021;39:6137-6143.

Knowledge about influenza and adherence to the recommendations for influenza vaccination of pregnant women after an educational intervention in Greece.

Maltezou HC, Pelopidas Koutroumanis P, Kritikopoulou C, Theodoridou K, Katerelos P, Tsiaousi I, Rodolakis A, Loutradis D.Hum Vaccin Immunother. 2019;15:1070-1074.

Author Response

Response to Reviewer 2 Comments

Point 1: Methods: line 57: please correct to “pregnant women” instead of “pregnancy”. 

Response 1:  Thank you, we have made corrections as suggested.

Point 2:  Methods: Do you know (approximately) the number of women participating on these social networks? If yes, please indicate the overall response rate (%) in the first line of Results. If no, please add this to the limitations paragraph (Discussion).  

Response 2: Unfortunately, we do not have access to such information. We added this to the limitations paragraph.

Point 3:  Table 1: You mention “studying”: you mean “university students”? if yes, please correct accordingly.

Response 3:  Yes, we meant university students. Corrected.

Point 4:  In Tables 3 and 4 there are significant associations with income. Please mention in Results section.

Response 4:  Thank you, we have made corrections as suggested.

Point 5:   Please add the following recently published articles in your reference list:

Vaccination programs for pregnant women in Europe, 2021. Maltezou HC, Effraimidou E, Cassimos DC, Medic S, Topalidou M, Konstantinidis T, Theodoridou M, Rodolakis A. Vaccine. 2021;39:6137-6143.

Knowledge about influenza and adherence to the recommendations for influenza vaccination of pregnant women after an educational intervention in Greece.

Maltezou HC, Pelopidas Koutroumanis P, Kritikopoulou C, Theodoridou K, Katerelos P, Tsiaousi I, Rodolakis A, Loutradis D.Hum Vaccin Immunother. 2019;15:1070-1074.

Response 5:  Thank you for your suggestion. We have added the articles.

Reviewer 3 Report

This paper presents the results of a cross-sectional survey of pregnant women regarding opinions on/attitudes towards influenza and COVID vaccination. This study is important in terms of identifying specific factors that can be addressed in efforts to increase vaccine uptake among this uniquely vulnerable population. While the data presented here are interesting, there are several serious questions that must be addressed prior to any publication. First, the authors do not include any reference to Institutional Review Board approval or exemption for this study. While it is an anonymous survey, it still qualifies as human subjects research and would require ethics approval. Without that ethics approval, the study should not be published.

If the authors do have this approval, that should be included in the Methods section.

Overall, the methods section should be expanded significantly, to include more details regarding the specific content of the survey (and total number of questions), how the survey was promoted/posted, whether any incomplete surveys were submitted, etc. The target number of completed surveys was not identified, and should be discussed. The program(s) used to conduct statistical analysis should be included in the methods section. Also unclear is whether participants were permitted to select more than one answer for the vaccine-knowledge (it is implied but never stated in the text).

In tables 3-8, it is not clear what this p-value represents; if this is done by chi-square, what is being used as the “expected” group? Chi-square may not be the optimal test to use in this instance, especially for tables 7 and 8 where there are multiple response choices and the data are divided into multiple demographic bins. In addition, some discussion of confounding (for education, location, and average income) may be beneficial, as these factors may not be independent of each other. 

For tables 5 and 6, the total number of responses should be included with the table title (as in previous tables).

The results presented in table 8 are quite interesting, but the formatting of the table precludes an easy understanding of the data presented. There may be an alternate way of displaying the data (heatmap or other visual display) that could demonstrate these results in a clearer manner.

The discussion should include an expanded section on limitations imposed by anonymous surveys. It also could include an expansion of the implications of these results particularly for vaccine distribution and uptake efforts.

The English is overall good, there are a few atypical constructions, but nothing that impairs understanding of the paper.

Author Response

Response to Reviewer 3 Comments

Point 1:   First, the authors do not include any reference to Institutional Review Board approval or exemption for this study. While it is an anonymous survey, it still qualifies as human subjects research and would require ethics approval. Without that ethics approval, the study should not be published.

If the authors do have this approval, that should be included in the Methods section.

Response 1:  According to the Polish law, our bioethics committee does not require approval for the survey work. The written information is sufficient and has been submitted.

Point 2:  Overall, the methods section should be expanded significantly, to include more details regarding the specific content of the survey (and total number of questions), how the survey was promoted/posted, whether any incomplete surveys were submitted, etc. The target number of completed surveys was not identified, and should be discussed. The program(s) used to conduct statistical analysis should be included in the methods section. Also unclear is whether participants were permitted to select more than one answer for the vaccine-knowledge (it is implied but never stated in the text).

Response 2:  Thank you for your suggestion. The methods section was expanded. We discussed and defined the target number of participants and included it in the methodology. We could also attach our questionnaire as an Appendix.

Point 3:  In tables 3-8, it is not clear what this p-value represents; if this is done by chi-square, what is being used as the “expected” group? Chi-square may not be the optimal test to use in this instance, especially for tables 7 and 8 where there are multiple response choices and the data are divided into multiple demographic bins. In addition, some discussion of confounding (for education, location, and average income) may be beneficial, as these factors may not be independent of each other.

Response 3:  Thank you very much for your suggestion. In our study, we focused on determining the distribution of categories in the groups, whether it is even or uneven and the Chi-square test was used for that.

Point 4:  For tables 5 and 6, the total number of responses should be included with the table title (as in previous tables).

Response 4:  Thank you, we have made corrections as suggested.

Point 5:  The results presented in table 8 are quite interesting, but the formatting of the table precludes an easy understanding of the data presented. There may be an alternate way of displaying the data (heatmap or other visual display) that could demonstrate these results in a clearer manner.

Response 5: We tried to provide this data in a different way, but we decided that the table would be the most appropriate and the clearest of all other possibilities.

Point 6:  The discussion should include an expanded section on limitations imposed by anonymous surveys. It also could include an expansion of the implications of these results particularly for vaccine distribution and uptake efforts.

Response 6: In our opinion, anonymity in research brings benefits rather than limitations. Anonymity can provide a sense of comfort and security for the respondents and thus contribute to a more open presentation of their views and feelings.

Round 2

Reviewer 2 Report

The authors successfully addressed my previous comments. 

Author Response

Thank you very much.

Reviewer 3 Report

The authors have added two clarifying graphs, which aid greatly in data presentation. The additions to tables 5 and 6 are also appreciated.

I appreciate the authors' clarification regarding bioethics approvals in Poland; a sentence stating that anonymous surveys do not require ethics committee approval should be added to the methods section, for readers that are less familiar with ethics approvals in different countries. Including the questionnaire as an appendix would provide a useful resource, and would be recommended.

I am still concerned regarding the statistical analysis in this paper. Chi-square is used for observed vs expected values, typically with a binary variable (see https://www.ncbi.nlm.nih.gov/pmc/articles/PMC8503070/ table 1 for an example); the multiple comparisons in tables 7 and 8 especially should not be analyzed with a single chi-square calculation. 

Author Response

Thank you very much for your opinion and suggestions. 

A sentence regarding bioethics approvals in Poland was added.

We have tried to improve our statistical analysis as much as possible. We used the univariate and multivariate logistic regression analyses for the categorical variables and completely revised Tables 3, 4 and 7, and used the Spearman rank coefficient in Table 8.